# Risk factors of phlebitis in patients admitted to the intensive care unit vary according to the duration of catheter dwelling: A post-hoc analysis of the AMOR-VENUS study

Yutaro Shinzato[1], Hideto Yasuda[1,2]\*, Takashi Moriya[1], Haruka Taira[1], Yuki Kishihara[1], Masahiro Kashiura[1], Yuki Kotani[3], Natsuki Kondo[4], Kosuke Sekine[5], Nobuaki Shime[6], Keita Morikane[7], on behalf of the AMOR-VENUS study group[¶]

1 Department of Emergency and Critical Care Medicine, Jichi Medical University Saitama Medical Center, Saitama, Japan, 2 Department of Clinical Research Education and Training Unit, Keio University Hospital Clinical and Translational Research Center (CTR), Tokyo, Japan, 3 Department of Intensive Care Medicine, Kameda Medical Center, Chiba, Japan, 4 Department of Emergency Medicine, Koga Community Hospital, Yaizu, Japan, 5 Department of Medical Engineer, Kameda Medical Center, Chiba, Japan, 6 Department of Emergency and Critical Care Medicine, Graduate School of Biomedical and Health Sciences, Hiroshima University, Hiroshima, Japan, 7 Division of Clinical Laboratory and Infection Control, Yamagata University Hospital, Yamagata, Japan

¶ Membership of the AMOR-VENUS study group is provided in the Acknowledgments.
\* yasudahideto@me.com, yasuda.hideto@keio.jp

## Abstract

### Aim

This study investigated the risk factors of peripheral intravenous catheter (PIVC)-related phlebitis in critically ill patients according to the duration of catheter dwelling.

### Methods

This was a post-hoc analysis of the AMOR-VENUS study involving 23 intensive care units (ICUs) in Japan. We included patients aged ≥ 18 admitted to the ICU and had PIVCs inserted during ICU admission. The primary outcome measure was phlebitis, and the risk factors of phlebitis were evaluated based on hazard ratios (HR) and 95% confidence intervals (CI). The duration of catheter dwelling was classified as (i) ≤ 24 h; (ii) > 24 h, ≤ 72 h; and (iii) > 72 h. Multivariable marginal Cox regression analysis was performed using the presumed risk factors for each group.

### Results

In total, 1,335 patients and 3,348 PIVCs were evaluated. Among patients with ≤ 24 h of catheter dwelling, phlebitis occurrence was associated with ICU admission for non-surgical management with ICU admission for elective surgery as the reference, standardized drug administration in the ICU, and dexmedetomidine administration in the ICU. Among those with > 24 h but ≤ 72 h of catheter dwelling, it was associated

**Data availability statement:** Data cannot be shared publicly due to the obtained approval, which stipulates that the data should not be disclosed unless necessary when registering the research in the registry. If the use of data is required for an important reason, the data will be provided by the corresponding author (yasu-dahideto@me.com or yasuda.hideto@keio.jp) or Cinical trial promotion departmentof Jichi Medical University Saitama Medical Center (s-suishin@jichi.ac.jp)to researchers who meet the criteria for access to confidential data.

**Funding:** The author(s) received no specific funding for this work.

**Competing interests:** The authors have declared that no competing interests exist.

with male sex with female sex as the reference, tetrafluoroethylene as the catheter material with polyurethane as the reference, nicardipine administration, and noradrenaline administration. Among those with > 72 h of catheter dwelling, it was associated with a catheter size ≥ 18 G and nicardipine administration.

## Conclusion

The risk factors for phlebitis varied with the duration of catheter dwelling. Individualized catheter management, considering the duration of catheter dwelling, may help avoid phlebitis in patients admitted to the ICU.

---

## Introduction

Peripheral intravascular catheters (PIVCs) are used in most intensive care unit (ICU) patients. However, PIVC insertion is associated with various adverse events such as hematoma, skin inflammation associated with drug leakage, and phlebitis [1]. Phlebitis occurs in 7.5% of critically ill patients with PIVCs [2]. Importantly, phlebitis may be considered a major complication since even mild phlebitis can cause pain and anxiety, and severe phlebitis can cause skin necrosis and infective endocarditis [3–5]. The risk factors for phlebitis in patients admitted to the ICU include having a body mass index (BMI) ≥ 30, catheter insertion by a physician, catheter insertion in the upper arm, and the administration of certain drugs, such as nicardipine and noradrenaline [6]. Previous research has shown that PIVC-related phlebitis can occur at different stages [6–8], suggesting that its onset may be affected by the catheter dwelling time. Furthermore, if the risk factors for phlebitis vary according to the duration of catheter dwelling, individualized catheter management may be recommended by considering these risk factors, particularly in critically ill patients with a high risk of phlebitis. We believe no previous study on the risk factors for PIVC-related phlebitis has stratified them by the duration of catheter dwelling. Therefore, this study aimed to investigate whether the risk factors for phlebitis vary with the duration of catheter dwelling.

## Materials and methods

### Study design and patients

This study was a post-hoc analysis of the AMOR-VENUS study, a prospective, multicenter cohort study conducted between January 1, 2018 and March 31, 2018 in 22 institutions and 23 ICUs in Japan [2]. The AMOR-VENUS has already received approval from the review board of the University Hospital Medical Information Network Clinical Trials Registry under the Japanese Clinical Trial Registry (registration number: UMIN000028019) and the need for informed consent was waived, and an opt-out recruitment method was employed. The need for a new ethical review was waived for this study because the approval for the AMOR-VENUS study included the post-hoc analysis. The manuscript has been prepared according to the Strengthening the Reporting of Observational Studies in Epidemiology (STROBE) guidelines [9] (Table in S1 File).

The AMOR-VENUS dataset included the data of patients aged ≥ 18 years who were admitted to the ICU and had PIVCs inserted during ICU admission. The detailed inclusion and exclusion criteria have been described previously (2). The current study excluded the following patients: (1) those with PIVCs inserted outside the ICU, (2) those with missing data on the duration of catheter dwelling, and (3) those in whom the catheter material was unclassifiable.

## Data collection

The following data were collected from the dataset: patient characteristics (age, sex, height, weight, BMI, Charlson Comorbidity Index [10], Acute Physiology and Chronic Health Evaluation (APACHE) II score [11], Simplified Acute Physiology Score II [12], Sequential Organ Failure Assessment score [13], ICU admission location, type of ICU admission, medical reasons for ICU admission, presence of sepsis at ICU admission, and requirement of mechanical ventilation), PIVC characteristics (the medical staff who inserted the catheter, whether standardized drug administration measures were followed in the ICU, insertion site, catheter materials, catheter size, antiseptic solution applied before catheterization, use of ultrasonography, number of attempts before successful insertion, difficulties during insertion, type of gloves, type of dressing, any infection during catheter dwelling, and duration of catheter dwelling), drugs administered via PIVCs during the ICU stay (ampicillin/sulbactam, dexmedetomidine, lipid emulsion, fentanyl, heparin, midazolam, nicardipine, and nor-adrenaline), ICU mortality, and outcome of phlebitis. Phlebitis was defined using the Phlebitis Scale developed by the Infusion Nurses Society [14]. Detailed information on the definition and methods of phlebitis evaluation has been described in the AMOR-VENUS study and in S2 File.

## Statistical analysis

The primary outcome measure was phlebitis, and the risk factors for phlebitis were analyzed based on the hazard ratios (HRs). Continuous variables were presented as means and standard deviations (SD) or as medians and interquartile ranges (IQRs) and were analyzed using analysis of variance or the Kruskal–Wallis test. These tests were performed to compare patient background characteristics between the three groups for assessing whether there were significant differences among the groups categorized by catheter insertion time. Meanwhile, categorical variables were presented as absolute counts and percentages (%) and were analyzed using the Fisher exact test or Pearson's chi-square test. These tests were used in the univariate analysis to identify differences among the groups based on catheter dwelling duration, rather than to examine associations with phlebitis. As a result, an initial understanding about background differences among the groups was provided. Subsequently, we applied multilevel Cox regression to account for hierarchical variability by incorporating patient and facility (ICU) outcomes as random effects. The duration of catheter dwelling was classified based on a previous study [15] and on the delimitations used clinically as follows: ≤ 24 h; > 24 h, ≤ 72 h; and > 72 h. Univariate and multivariable marginal Cox regression analyses were used for each patient and institution classifications stratified by the duration of catheter dwelling. These analyses assessed the association between the timing of phlebitis onset and presumed risk factors, as there were within-patient and within-institution correlations between the PIVC characteristics. The endpoint of this study was not the incidence of phlebitis but the time from catheter insertion to phlebitis onset. The time to phlebitis was estimated based on 4-hour monitoring intervals. In the marginal COX regression model, the starting point (time zero) was established as the moment the PIVC was inserted in the ICU. The model was censored at the point of PIVC removal or when a patient was discharged from the ICU with the PIVC still in place. Estimated risk factors for phlebitis, extracted based on a previous study [6], have been described in S3 File. Since missing data were randomly distributed, imputation was not performed, and only patients with complete data were included in the analysis. Effect estimates were described using HRs and 95% confidence intervals (CI). All statistical analyses were performed using EZR version 1.38 (Saitama Medical Center, Jichi Medical University, Saitama, Japan) and SAS Studio (SAS Inc., Cary, NC), and a two-sided $p$-value of < 0.05 was considered statistically significant.

## Results

### Patient characteristics

In total, 2,741 patients with 7,118 PIVCs were included in the analysis (Fig 1). Among them, 1,406 patients and 3,770 PIVCs were excluded because of catheter insertion outside the ICU (n = 1,382 patients and 3,689 PIVCs), missing data on the duration of catheter dwelling (n = 10 patients and 13 catheters), and the use of unclassifiable catheter materials (n = 77 patients and 335 catheters). Finally, 491 patients (36.8%) and 1,040 catheters (31.1%) were categorized into the ≤24 h group; 498 patients (37.3%) and 1,324 catheters (39.5%), the >24 h, ≤72 h group; and 346 patients (25.9%) and 984 catheters (29.4%), the >72 h group. Missing data are described in Tables 1 and 2. The characteristics of the included patients are presented in Table 1. Overall, the median (IQR(age was 71.0 (18.0) years; 842 (63.1%) patients were men, 484 (36.3%) patients were admitted to the ICU due to cardiogenic disease, and 681 (51.0%) patients required mechanical ventilation within 24 h of ICU admission. Phlebitis during ICU admission occurred in 99 patients, yielding an occurrence rate of 7.4%. There were significant differences in almost all the variables among the groups stratified by the duration of catheter dwelling.

### PIVC characteristics

The characteristics of the PIVCs are listed in Table 2. Overall, 3,300 catheters (98.6%) were inserted in accordance with standardized drug administration measures: 2,369 catheters (70.8%) were inserted by nurses, 1,802 catheters (53.8%) were inserted in the forearm, and 1,252 (38.6%) catheters were made of tetrafluoroethylene. The median (IQR) duration of catheter dwelling was 46.2 h (21.5–82.7 h), and fentanyl was the most commonly administered drug (n = 451 patients, 16.6%). Almost all variables showed significant differences among the groups stratified by the duration of catheter dwelling.

### Risk factors for phlebitis by the duration of catheter dwelling

Univariate and multivariate marginal Cox regression analyses were performed for each group stratified by the duration of catheter dwelling (Table 3 and Table in S4 File). Regarding the multivariable marginal Cox regression analysis, the

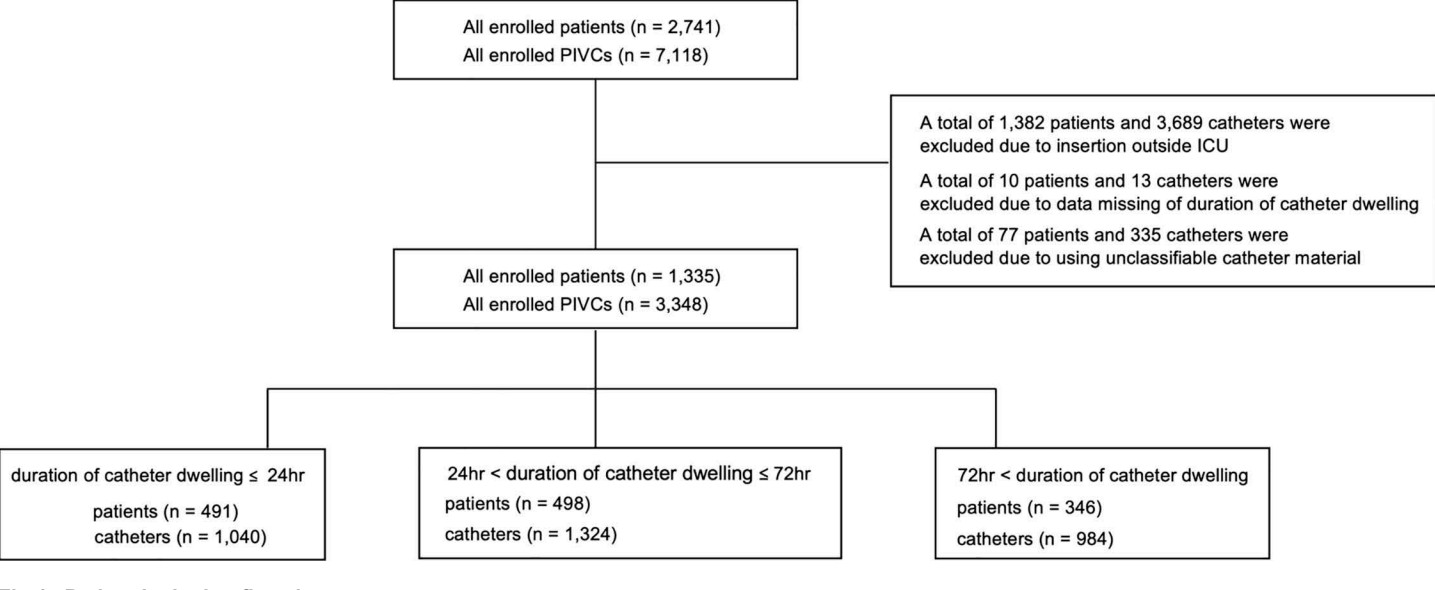

**Fig 1. Patient inclusion flowchart.**

**Table 1. Patient characteristics at ICU admission.**

| Variables | Total | Duration of catheter dwelling groups | | | |
|---|---|---|---|---|---|
| | n = 1,335 | ≤ 24 h<br>n = 491<br>(36.8%) | >24 h, ≤72 h<br>n = 498<br>(37.3%) | > 72 h<br>n = 346<br>(25.9%) | p-value |
| Age (years) | 71.0 {18.0} | 70.0 {19.0} | 72.5 {18.0} | 70.0 {17.0} | 0.63 |
| Male sex | 842 (63.1) | 322 (65.6) | 303 (60.8) | 217 (62.7) | 0.3 |
| Body height[a] (cm) | 160.8 [9.8] | 161.3 [9.6] | 160.2 [9.9] | 160.8 [10.0] | 0.22 |
| Body weight[b] (kg) | 58.0 [18.0] | 59.0 [19.0] | 59.0 [19.0] | 58.0 [17.0] | 0.14 |
| BMI[b] | 22.4 [5.3] | 22.5 [4.9] | 22.8 (5.6) | 22.1 [5.0] | 0.07 |
| APACHE II score[c] | 17.0 {11.0} | 14.0 {9.0} | 17.0 {10.0} | 19.0 {11.0} | <.01 |
| SAPS II score[c] | 40.0 {27.0} | 31.0 {26.0} | 40.0 {24.0} | 47.0 {24.0} | <.01 |
| SOFA score [c] | 6.0 {6.0} | 4.0 {5.0} | 6.0 {4.0} | 8.0 {5.0} | <.01 |
| Charlson Comorbidity Index | 4.0 {3.0} | 4.0 {3.0} | 4.0 {3.0} | 4.0 {3.0} | 0.61 |
| ICU admission from | | | | | |
| Operating room | 597 (44.7) | 273 (55.6) | 192 (38.6) | 132 (38.2) | <.01 |
| Emergency room | 506 (37.90) | 163 (33.2) | 227 (45.6) | 116 (33.5) | <.01 |
| General ward | 177 (13.26) | 43 (8.8) | 61 (12.2) | 73 (21.1) | 0.02 |
| Outpatients | 6 (0.45) | 2 (0.4) | 2 (0.4) | 2 (0.6) | 1 |
| Transfer from other hospital | 49 (3.67) | 10 (2.0) | 16 (3.2) | 23 (6.6) | 0.07 |
| Type of admission to ICU | | | | | |
| Elective surgical | 376 (28.1) | 206 (42.0) | 120 (24.1) | 50 (14.5) | <.01 |
| Emergency surgical | 221 (16.6) | 67 (13.6) | 72 (14.5) | 82 (23.7) | 0.45 |
| Non-surgical | 738 (55.3) | 218 (44.4) | 306 (61.4) | 214 (61.8) | <.01 |
| Medical reason for ICU admission | | | | | |
| Cardiology | 484 (36.25) | 151 (30.8) | 208 (41.8) | 125 (36.1) | <.01 |
| Pulmonary | 182 (13.63) | 58 (11.8) | 56 (11.2) | 68 (19.7) | 0.51 |
| Gastrointestinal | 184 (13.78) | 90 (18.3) | 61 (12.2) | 33 (9.5) | <.01 |
| Neurology | 195 (14.61) | 80 (16.3) | 68 (13.7) | 47 (13.6) | 0.014 |
| Trauma | 52 (3.9) | 14 (2.9) | 21 (4.2) | 17 (4.9) | 0.49 |
| Urology | 16 (1.20) | 11 (2.2) | 4 (0.8) | 1 (0.3) | <.01 |
| Gynecology | 12 (0.90) | 9 (1.8) | 2 (0.4) | 1 (0.3) | <.01 |
| Skin/tissue | 23 (1.72) | 6 (1.2) | 9 (1.8) | 8 (2.3) | 0.74 |
| Others | 66 (4.94) | 35 (7.1) | 21 (4.2) | 10 (2.9) | <.01 |
| Sepsis at ICU admission | | | | | |
| Sepsis | 90 (9.96) | 23 (4.7) | 29 (5.8) | 38 (11.0) | 0.15 |
| Septic shock | 133 (9.96) | 31 (6.3) | 46 (9.2) | 56 (16.2) | 0.03 |
| Mechanical ventilation or | 681 (51.01) | 332 (67.6) | 250 (50.2) | 99 (28.6) | <.01 |
| Not[c] | 654 (48.99) | 159 (32.4) | 248 (49.8) | 247 (71.4) | <.01 |
| Phlebitis | 99 (7.4) | 31 (9.9) | 52 (11.4) | 16 (5.0) | <.01 |

Missing data: a, n = 2 (0.15%); b, n = 4 (0.3%); c, n = 90 (6.74%).

Data are presented as n (%) or as the mean [SD] or as the median {IQR}.

APACHE, Acute Physiology, and Chronic Health Evaluation; BMI, body mass index; ICU, intensive care unit; PIVC, peripheral intravascular catheter; SD, standard deviation.

**Table 2. PIVC characteristics during insertion.**

| Variables | Total | Duration of catheter dwelling groups | | | |
|---|---|---|---|---|---|
| | n = 3,348 | ≤ 24 h<br>n = 1,040<br>(31.1%) | > 24 h, ≤ 72 h<br>n = 1,324<br>(39.5%) | > 72 h<br>n = 984<br>(29.4%) | p-value |
| Medical staff inserting the catheter[a] | | | | | |
| Doctor | 283 (8.5) | 71 (9.0) | 93 (8.7) | 119 (15.0) | <.01 |
| Nurse | 2369 (70.8) | 716 (90.9) | 977 (91.3) | 676 (85.0) | <.01 |
| Medical technologist | 1 (0.03) | 1 (0.1) | 0 (0) | 0 (0) | -‡ |
| Provision of standardized drug administration measures in the ICU | 3,300 (98.6) | 1,024 (98.5) | 1,302 (98.3) | 974 (99.0) | 0.40 |
| Insertion site[b] | | | | | |
| Forearm | 1,802 (53.8) | 576 (55.7) | 730 (55.6) | 496 (50.8) | <.01 |
| Upper arm | 352 (10.5) | 112 (10.8) | 140 (10.7) | 100 (10.2) | 0.03 |
| Elbow | 161 (4.8) | 49 (4.7) | 60 (4.6) | 52 (5.3) | 0.54 |
| Wrist | 160 (4.8) | 57 (5.5) | 61 (4.6) | 42 (4.3) | 0.15 |
| Hand | 495 (14.8) | 159 (15.4) | 179 (13.6) | 157 (16.1) | 0.41 |
| Lower leg | 219 (6.5) | 52 (5.0) | 90 (6.9) | 77 (7.9) | <.01 |
| Dorsal foot | 136 (4.1) | 30 (2.9) | 53 (4.0) | 53 (5.4) | 0.02 |
| Catheter material | | | | | |
| PEU-Vialon°† | 1080 (32.3) | 297 (28.6) | 408 (30.8) | 375 (38.1) | <.01 |
| Polyethylene | 976 (29.2) | 291 (28.0) | 388 (29.3) | 297 (30.2) | <.01 |
| Tetrafluoroethylene | 1292 (38.6) | 452 (43.5) | 528 (39.9) | 312 (31.7) | <.01 |
| Catheter size[c] | | | | | |
| 14G | 1 (0.03) | 0 (0) | 1 (0.1) | 0 (0) | -‡ |
| 16G | 72 (2.1) | 21 (2.0) | 15 (1.2) | 35 (3.6) | 0.01 |
| 18G | 88 (2.6) | 44 (4.3) | 26 (2.0) | 18 (1.9) | <.01 |
| 20G | 868 (25.9) | 258 (25.1) | 332 (25.5) | 278 (28.6) | <.01 |
| 22G | 2,211 (66.0) | 686 (66.7) | 903 (69.5) | 622 (64.0) | <.01 |
| 24G | 61 (1.8) | 19 (1.8) | 23 (1.8) | 19 (2.0) | 0.77 |
| Antiseptic solution before catheterization[d] | | | | | |
| None | 8 (0.2) | 1 (0.1) | 3 (0.3) | 4 (0.5) | 0.42 |
| Alcohol | 2572 (76.8) | 768 (98.0) | 1048 (97.9) | 756 (96.7) | <.01 |
| 0.2% chlorhexidine alcohol | 20 (0.6) | 5 (0.6) | 8 (0.7) | 7 (0.9) | 0.71 |
| 0.5% chlorhexidine alcohol | 15 (0.5) | 5 (0.6) | 4 (0.4) | 6 (0.8) | 0.82 |
| 1.0% chlorhexidine alcohol | 17 (0.5) | 4 (0.5) | 5 (0.5) | 8 (1.0) | 0.47 |
| 10% povidone iodine | 2 (0.06) | 0 (0) | 2 (0.2) | 0 (0) | -‡ |
| Other | 3 (0.09) | 1 (0.1) | 1 (0.1) | 1 (0.1) | 1.0 |
| Use of ultrasonography[e] | 58 (1.7) | 10 (1.3) | 19 (1.8) | 29 (3.8) | <.01 |
| Number of attempts for insertion[f] | | | | | |
| 1 | 2098 (62.7) | 634 (82.0) | 836 (79.5) | 628 (81.8) | <.01 |
| 2 | 309 (9.2) | 84 (10.9) | 141 (13.4) | 84 (10.9) | <.01 |
| 3 | 130 (3.9) | 38 (4.9) | 56 (5.3) | 36 (4.7) | 0.06 |
| 4 | 25 (0.8) | 7 (0.9) | 8 (0.8) | 10 (1.3) | 0.76 |
| 5 | 15 (0.5) | 4 (0.5) | 7 (0.7) | 4 (0.5) | 0.55 |
| ≧ 6 | 16 (0.5) | 6 (0.8) | 4 (0.4) | 6 (0.8) | 0.78 |
| Difficulties with the insertions[g] | | | | | |
| Easy | 1221 (36.5) | 381 (49.9) | 492 (47.2) | 348 (45.5) | <.01 |
| Slightly easy | 768 (22.9) | 211 (27.6) | 313 (30.0) | 244 (31.9) | <.01 |

*(Continued)*

**Table 2.** (Continued)

| Variables | Total | Duration of catheter dwelling groups | | | |
|---|---|---|---|---|---|
| | n = 3,348 | ≤ 24 h n = 1,040 (31.1%) | > 24 h, ≤ 72 h n = 1,324 (39.5%) | > 72 h n = 984 (29.4%) | *p*-value |
| Slightly difficult | 449 (13.4) | 127 (16.6) | 189 (18.1) | 133 (17.4) | <.01 |
| Difficult | 133 (4.0) | 45 (5.9) | 49 (4.7) | 39 (5.1) | 0.57 |
| Glove[h] | | | | | |
| Sterile | 18 (0.5) | 3 (0.4) | 11 (1.0) | 4 (0.5) | 0.04 |
| Non-sterile | 2471 (73.8) | 748 (96.3) | 999 (94.4) | 724 (94.1) | <.01 |
| None | 115 (3.4) | 26 (3.3) | 48 (4.5) | 41 (5.3) | 0.04 |
| Dressing[i] | | | | | |
| Chlorhexidine-impregnated dressing | 0 (0) | 0 (0) | 0 (0) | 0 (0) | -‡ |
| Sterile polyurethane dressing | 3266 (97.6) | 1021 (98.7) | 1288 (97.7) | 957 (97.7) | 0.03 |
| Non-sterile polyurethane dressing | 57 (1.7) | 10 (1.0) | 26 (2.0) | 21 (2.1) | <.01 |
| Gauze dressing | 1 (0.03) | 0 (0) | 1 (0.1) | 0 (0) | -‡ |
| Tape dressing | 8 (0.2) | 3 (0.3) | 3 (0.3) | 2 (0.2) | 0.88 |
| Any infection during catheter dwelling | 792 (23.7) | 199 (19.1) | 296 (22.4) | 297 (30.2) | <.01 |
| Duration of catheter dwelling (hours) | 46.2 [21.5–82.7] | 15.9 [7.8–20.8] | 46.3 [34.4–61.0] | 114.4 [88.6–150.9] | <.01 |
| Administered drug | | | | | |
| Ampicillin/sulbactam | 172 (5.4) | 45 (4.3) | 76 (5.7) | 74 (7.5) | <.01 |
| Dexmedetomidine | 281 (8.4) | 42 (4.0) | 111 (8.4) | 128 (13.0) | <.01 |
| Lipid emulsion | 300 (9.8) | 37 (3.6) | 116 (8.8) | 147 (14.9) | <.01 |
| Fentanyl | 451 (16.6) | 80 (7.7) | 171 (12.9) | 200 (20.3) | <.01 |
| Heparin | 315 (9.4) | 60 (5.8) | 98 (7.4) | 157 (16.0) | <.01 |
| Midazolam | 56 (1.7) | 11 (1.1) | 20 (1.5) | 25 (2.5) | 0.03 |
| Nicardipine | 300 (9.0) | 86 (8.3) | 120 (9.1) | 94 (9.6) | 0.60 |
| Noradrenaline | 86 (2.6) | 23 (2.2) | 40 (3.0) | 23 (2.3) | 0.40 |
| Phlebitis | 303 (9.1) | 103 (9.9) | 151 (11.4) | 49 (5.0) | <.01 |

Missing data: a, n = 695 (20.8%); b, n = 23 (0.69%); c, n = 48 (1.4%); d, n = 711 (21.2%); e, n = 738 (22.0%); f, n = 755 (22.6%); g, n = 777 (23.2%); h, n = 744 (22.2%); i, n = 16 (0.5%)

†PEU-Vialon® is polyurethane.

‡This value could not be calculated.

Data are presented as n (%) or as the median [IQR].

Abbreviations: APACHE, Acute Physiology and Chronic Health Evaluation; ICU, intensive care unit; IQR, interquartile range; PIVC, peripheral intravascular catheter; SD, standard deviation

number of patients actually included in the groups was 732/1,040 catheters; 1,021/1,324 catheters; and 771/984 catheters in the ≤ 24h; > 24h, ≤ 72h; and > 72h groups, respectively. Multivariable marginal Cox regression analysis showed that the risk of phlebitis increased when the following factors were present: (i) in the ≤ 24h catheter dwelling group, ICU admission for non-surgical management with elective surgery as the reference (HR [95% CI]: 2.8 [1.34–5.85], *p* < 0.01), standardized drug administration in the ICU (HR [95% CI]: 0.27 [0.08–0.84], *p* = 0.02), and dexmedetomidine administration (HR [95% CI]: 2.82 [1.39–5.68], *p* < 0. 01); (ii) in the > 24h, ≤ 72h group: male sex with female sex as the reference (HR [95% CI]: 0.67 [0.46–0.98], *p* = 0.04), tetrafluoroethylene as the catheter material with polyurethane as the reference (HR [95% CI]: 0.54 [0.35–0.84], *p* < 0.01), nicardipine administration (HR [95% CI]: 2.41 [1.54–3.8], *p* < 0.01), and noradrenaline administration (HR [95% CI]: 4.12 [2.08–8.15], *p* < 0.01); and (iii) in the > 72h group: catheter size ≥ 18G (HR [95% CI]: 11.02 [1.91–63.52], *p* < 0.01) and nicardipine administration (HR [95% CI]: 3.45 [1.36–8.79], *p* < 0.01) (Table 3).

**Table 3. Multivariable analysis with marginal Cox regression analysis for the risk factors of phlebitis stratified by the duration of catheter dwelling.**

| Variables | Duration of catheter dwelling groups | | | | | |
|---|---|---|---|---|---|---|
| | ≤ 24 h (n = 732/1,040) Phlebitis: n = 103 (9.9%) | | > 24 h, ≤ 72 h (n = 1,021/1,324) Phlebitis: n = 151 (11.4%) | | > 72 h (n = 771/984) Phlebitis: n = 49 (5.0%) | |
| | HR (95% CI) | *p*-value | HR (95% CI) | *p*-value | HR (95% CI) | *p*-value |
| Age (years) | 1.01 (1.0–1.03) | 0.16 | 1.0 (0.99–1.02) | 0.72 | 1.02 (0.99–1.05) | 0.16 |
| Male sex | 1.07 (0.66–1.72) | 0.79 | 0.67 (0.46–0.98) | 0.04 | 0.79 (0.38–1.64) | 0.53 |
| BMI (kg/m$^2$) | | | | | | |
| 18.6–25 | ref | – | ref | – | ref | – |
| ≤ 18.5 | 1.13 (0.58–2.23) | 0.72 | 1.19 (0.71–2.0) | 0.52 | 1.6 (0.63–4.09) | 0.33 |
| > 25 < | 0.72 (0.41–1.24) | 0.23 | 0.96 (0.63–1.46) | 0.84 | 1.53 (0.68–3.43) | 0.3 |
| APACHE II score | | | | | | |
| 16–25 | ref | – | ref | – | ref | – |
| ≤ 15 | 0.87 (0.5–1.53) | 0.63 | 1.01 (0.67–1.51) | 0.96 | 1.5 (0.67–3.38) | 0.33 |
| ≥ 26 | 0.94 (0.52–1.7) | 0.83 | 0.69 (0.4–1.18) | 0.17 | 0.34 (0.1–1.09) | 0.07 |
| Type of admission to ICU | | | | | | |
| Elective surgical | Ref | – | ref | – | ref | – |
| Emergency surgical | 1.01 (1.0–1.03) | 0.16 | 1.08 (0.49–2.4) | 0.85 | 0.94 (0.25–3.51) | 0.92 |
| Non-surgical | 2.8 (1.34–5.85) | <.01 | 1.21 (0.6–2.45) | 0.6 | 1.15 (0.36–3.72) | 0.81 |
| Provision of standardized drug administration measures in the ICU | 0.27 (0.08–0.84) | 0.02 | 1.07 (0.35–3.26) | 0.91 | 0.36 (0.04–3.43) | 0.37 |
| Medical staff inserting the catheter | | | | | | |
| Nurse | ref | – | ref | – | ref | – |
| Doctor | 0.93 (0.43–2.02) | 0.85 | 0.79 (0.36–1.75) | 0.56 | 0.35 (0.11–1.1) | 0.07 |
| Insertion site | | | | | | |
| Forearm | ref | – | ref | – | ref | – |
| Upper arm | 0.41 (0.17–1.01) | 0.05 | 0.55 (0.29–1.04) | 0.07 | 0.63 (0.17–2.35) | 0.49 |
| Elbow | 1.58 (0.63–3.95) | 0.33 | 0.29 (0.07–1.21) | 0.09 | 1.44 (0.37–5.6) | 0.6 |
| Wrist | 0.59 (0.18–1.96) | 0.39 | 0.3 (0.07–1.27) | 0.1 | 1.4 (0.29–6.85) | 0.67 |
| Hand | 0.81 (0.39–1.68) | 0.57 | 0.54 (0.28–1.05) | 0.07 | 1.3 (0.47–3.51) | 0.63 |
| Lower leg | 1.37 (0.65–2.88) | 0.4 | 0.58 (0.3–1.13) | 0.11 | 0.44 (0.06–3.38) | 0.43 |
| Dorsal foot | 1.17 (0.34–4.08) | 0.81 | 1.41 (0.69–2.87) | 0.34 | 2.15 (0.59–7.86) | 0.25 |
| Catheter material | | | | | | |
| Polyurethane | ref | – | ref | – | ref | – |
| PEU-Vialon˚† | 1.45 (0.75–2.8) | 0.27 | 0.7 (0.43–1.13) | 0.14 | 0.8 (0.32–2.05) | 0.65 |
| Tetrafluoroethylene | 1.31 (0.73–2.37) | 0.37 | 0.54 (0.35–0.84) | <.01 | 1.25 (0.45–3.47) | 0.66 |
| Catheter size | | | | | | |
| 22–24G | Ref | – | ref | – | ref | – |
| ≥ 18G | 2.03 (0.41–10.18) | 0.39 | 1.25 (0.16–9.58) | 0.83 | 11.02 (1.91–63.52) | <.01 |
| 20G | 0.63 (0.33–1.21) | 0.17 | 0.64 (0.36–1.12) | 0.12 | 1.74 (0.76–3.99) | 0.19 |
| Administered drug | | | | | | |
| Ampicillin/sulbactam | 0.4 (0.15–1.03) | 0.06 | 1.66 (0.88–3.15) | 0.12 | 0.88 (0.2–3.91) | 0.87 |
| Dexmedetomidine | 2.82 (1.39–5.68) | <.01 | 1.43 (0.81–2.53) | 0.22 | 0.57 (0.18–1.81) | 0.34 |
| Lipid emulsion | 1.8 (0.72–4.5) | 0.21 | 1.01 (0.5–2.05) | 0.98 | 0.98 (0.34–2.83) | 0.97 |
| Fentanyl | 1.88 (0.93–3.79) | 0.08 | 0.83 (0.47–1.47) | 0.53 | 0.55 (0.18–1.74) | 0.31 |

*(Continued)*

**Table 3.** (Continued)

| Variables | Duration of catheter dwelling groups | | | | | |
|---|---|---|---|---|---|---|
| | ≤ 24 h (n = 732/1,040) Phlebitis: n = 103 (9.9%) | | > 24 h, ≤ 72 h (n = 1,021/1,324) Phlebitis: n = 151 (11.4%) | | > 72 h (n = 771/984) Phlebitis: n = 49 (5.0%) | |
| | HR (95% CI) | p-value | HR (95% CI) | p-value | HR (95% CI) | p-value |
| Heparin | 1.91 (0.75–4.86) | 0.17 | 0.84 (0.38–1.86) | 0.67 | 1.15 (0.44–2.97) | 0.78 |
| Midazolam | 1.1 (0.3–4.04) | 0.88 | 1.95 (0.66–5.73) | 0.23 | 1.58 (0.25–9.83) | 0.63 |
| Nicardipine | 0.84 (0.37–1.92) | 0.68 | 2.41 (1.54–3.8) | <.01 | 3.45 (1.36–8.79) | <.01 |
| Noradrenaline | 2.24 (0.7–7.21) | 0.18 | 4.12 (2.08–8.15) | <.01 | 1.48 (0.28–7.87) | 0.65 |

† PEU-Vialon° is specified polyurethane.

Abbreviations: APACHE, Acute Physiology, and Chronic Health Evaluation; BMI, body mass index; CI, confidence interval; ER, emergency room; ICU, intensive care unit; IQR, interquartile range; HR, hazard ratio; PIVC, peripheral intravenous catheter

## Discussion

### Main findings

The current study found that the risk factors for phlebitis differed according to the duration of catheter dwelling (≤ 24 h, > 24 h, ≤ 72 h, and > 72 h). There are several possible reasons for these results in each group, which are discussed below.

### Catheter size

The Infusion Nurse Society recommends using the smallest catheter possible because catheters larger than 20G are more likely to cause phlebitis than catheters smaller than 20G [16]. The risk of phlebitis when using a large catheter is attributed to mechanical stimulation of the vessel wall by the catheter, which can cause vasculitis and subcutaneous edema [17, 18]. A large catheter size relative to the vessel diameter also causes phlebitis [6,19]. Therefore, we speculate that a large catheter that comes in contact with the vessel wall may hasten the occurrence of phlebitis. In this study, catheter size ≥ 18G was a risk factor for phlebitis in the ≥ 72 h group. This could be because, compared with the other groups, this group included more patients with septic shock, in whom catheters of ≥ 18G are commonly used, and a larger catheter size was used for more extended period for resuscitation.

### Drugs

The study on which we based the criteria for drug selection [6] showed that the drugs commonly administered in the ICU, such as nicardipine and noradrenaline, are risk factors for phlebitis. Consistently, the current study found that the use of dexmedetomidine, nicardipine, and noradrenaline increased the risk of phlebitis. Previous studies have shown that the dose, duration, and rate of administration of drugs [8,16,20] are factors in drug-induced phlebitis and may have been associated with the occurrence of phlebitis in patients receiving the abovementioned drugs in this study.

### Occurrence of phlebitis

We found that phlebitis occurred more frequently within 72 h than after 72 h of ICU admission. This may be because the current study was on patients admitted to the ICU who were treated with many drugs and devices in the ICU due to their critically ill condition. Therefore, they may have had many phlebitis risk factors. In our previous report [6], we found discrepant findings, where the duration of catheter dwelling was shorter in patients with phlebitis compared with those without phlebitis (Duration of catheter dwell, median (IQR), hours: 37.0 (19.2–57.6) in patients with phlebitis vs. 44.8 (21.0–81.5)

in those without phlebitis, $p<0.01$). The median duration of catheter dwelling in patients without phlebitis also fell within the 24–72 h range. This likely reflected that the period with higher phlebitis incidence included patients in the later phase, specifically after 72 h, when there were fewer risk factors. Therefore, we suggest that categorizing by the duration of catheter dwelling and demonstrating the relationship with associated risk factors could provide meaningful insights. Considering that PIVC-related phlebitis increases 72–96 h after insertion [7,8,15], the U.S. Centers for Disease Control and Prevention guidelines [21] recommend replacing PIVCs after 72–96 h. However, these studies included many mildly ill patients; thus, the recommendation may not apply to critically ill patients admitted to the ICU. For patients with risk factors for phlebitis admitted to the ICU, we suggest that clinicians consider changing or removing PIVCs, even if the PIVCs have been in place for less than 72 h.

## Implications

Our findings suggest that the risk factors associated with phlebitis could vary according to the duration of catheter dwelling (≤ 24 h, > 24 h, ≤ 72 h, and > 72 h). This variability highlights the potential benefit of adjusting catheter replacement timing based on the duration of catheter dwelling and specific patient risk factors. For instance, in patients who exhibit risk factors for phlebitis within the first 24 h, scheduled replacement within this period might reduce phlebitis incidence. Conversely, if no significant risk factors are evident within the first 72 h but emerge after this period, a monitoring strategy up to 72 h might be appropriate, followed by catheter replacement as necessary. This approach emphasizes the importance for dynamic monitoring strategies for reassessing catheter-related complications at critical time points rather than following a fixed replacement schedule. Implementing a tailored catheter management strategy considering both individual risk factors and catheter dwell time, healthcare providers can optimize catheter replacement timing. Such an approach could lead to the development of practical guidelines for catheter management in critically ill patients, balancing the need for minimizing complications while avoiding unnecessary catheter replacements. Ultimately, this strategy might improve patient outcomes and reduce the complications associated with prolonged catheter use.

## Limitations

This study has certain limitations. First, we divided the patients into three groups according to the duration of catheter dwelling based on groupings made in similar studies; however, there could be different interpretations of the group timings. For example, similar studies have used 72 h [7,15]; however, we also considered 24 h to be one of the delimitations for clinical use, leading to the 3-group system used. The results might have differed if the analyses were conducted in larger or smaller groups. Second, there were cases wherein catheters were early removed due to suspected phlebitis, leading to shortened duration of catheter dwelling. This reduction in the duration of catheter dwelling might have impacted the relationship between phlebitis incidence and its associated risk factors, as the longer duration of catheter dwelling could potentially yield different outcomes considering phlebitis occurrence. Due to limitations of evaluating this effect comprehensively, further investigation is warranted to fully understand the impact of shortened duration of catheter dwelling on phlebitis risk. Finaly, in the multivariate multilevel marginal Cox regression analysis, the drug used was treated as a binary variable, which may have underestimated the risk associated with using that drug and failed to detect statistically significant differences. The effect of a drug depends not only on its administration but also on its dose, concentration, and rate of administration. Since both high and low dosages of the drug were treated as a binary variable 'administered' within the same category, without differentiation, the effect of the drug may have been underestimated in this current study. Therefore, we suggest that future studies treat administered drugs as continuous variables to reach more specific conclusions.

## Conclusions

The risk factors for phlebitis varied with the duration of catheter dwelling. Individualized catheter management regimes should consider the duration of catheter dwelling to avoid phlebitis in patients admitted to the ICU.

## Supporting information

**S1 File. STROBE Statement—checklist of items that should be included in the reports of observational studies.**
(DOCX)

**S2 File. Definition of phlebitis according to the Infusion Nurses Society (INS).**
(DOCX)

**S3 File. Presumed risk factors for phlebitis.**
(DOCX)

**S4 File. Univariate and multivariable analysis for adverse events using logistic regression analysis.**
(DOCX)

## Acknowledgments

We would like to acknowledge Yoshiro Hayashi, Ryohei Yamamoto, Toru Takebayashi, Mikihiro Maeda, Takuya Shiga, Taku Furukawa, Mototaka Inaba, Sachito Fukuda, Kiyoyasu Kurahashi, Sarah Murakami, Yusuke Yasumoto, Tetsuro Kamo, Masaaki Sakuraya, Rintaro Yano, Toru Hifumi, Masahito Horiguchi, Izumi Nakayama, Masaki Nakane, Kohei Ota, Tomoaki Yatabe, Masataka Yoshida, Maki Murata, Kenichiro Fujii, Junki Ishii, Yui Tanimoto, Toru Takase, Tomoyuki Masuyama, Masamitsu Sanui, Takuya Kawaguchi, Junji Kumasawa, Norimichi Uenishi, Toshihide Tsujimoto, Kazuto Ono-zuka, Shodai Yoshihiro, Takakiyo Tatumichi, Akihiro Inoue, Bun Aoyama, Moemi Okazaki, Takuya Fujimine, Jun Suzuki, Tadashi Kikuchi, Satomi Tone, Mariko Yonemori, Kenji Nagaoka, Naomi Kitano, Masaki Ano, Ichiro Nakachi, Ai Ishimoto, Misa Torii, Junichi Maehara, Yasuhiro Gushima, Noriko Iwamuro, and the registered nurses of the ICU of IUHW Mita Hospital for their support with data collection at 22 institutions (Kameda Medical Center, Hiroshima University Hospital, Jichi Medical University Saitama Medical Center, Japanese Red Cross Musashino Hospital, Sakai City Medical Center, Fujita Health Univresity, Japanese Red Cross Society Wakayama Medical Center, JA Hiroshima General Hospital, Kagawa University Hospital, Kochi Medical School Hospital, Japanese Red Cross Kyoto Daiichi Hospital, Tohoku University Hospital, Nerima Hikarigaoka Hospital, Saiseikai Kumamoto Hospital, Okinawa Chubu Hospital, Shiroyama Hospital, Okayama Saiseikai General Hospital, Nagasaki University Hospital, Saiseikai Utsunomiya Hospital, Mitsui Memorial Hospital, International University of Health and Welfare Mita Hospital, and Yamagata University Hospital). The group members' names will be searchable using their individual PubMed records. The authors acknowledge the support of Editage (www.editage. jp) for the English language editing.

## Author contributions

**Conceptualization:** Yutaro Shinzato, Hideto Yasuda, Takashi Moriya, Haruka Taira, Yuki Kishihara, Masahiro Kashiura, Yuki Kotani, Natsuki Kondo, Kosuke Sekine, Nobuaki Shime, Keita Morikane.

**Data curation:** Yutaro Shinzato, Hideto Yasuda, Yuki Kishihara, Yuki Kotani, Natsuki Kondo, Kosuke Sekine.

**Formal analysis:** Yutaro Shinzato, Hideto Yasuda, Masahiro Kashiura.

**Funding acquisition:** Hideto Yasuda.

**Investigation:** Yutaro Shinzato, Hideto Yasuda, Yuki Kishihara, Masahiro Kashiura, Yuki Kotani, Natsuki Kondo, Kosuke Sekine.

**Methodology:** Yutaro Shinzato, Hideto Yasuda, Takashi Moriya, Masahiro Kashiura, Nobuaki Shime, Keita Morikane.

**Project administration:** Yutaro Shinzato, Hideto Yasuda, Takashi Moriya, Yuki Kishihara, Masahiro Kashiura, Yuki Kotani, Natsuki Kondo, Kosuke Sekine, Nobuaki Shime, Keita Morikane.

**Resources:** Yutaro Shinzato, Hideto Yasuda, Yuki Kishihara, Masahiro Kashiura, Yuki Kotani, Natsuki Kondo, Kosuke Sekine.

**Software:** Yutaro Shinzato, Hideto Yasuda, Masahiro Kashiura.

**Supervision:** Yutaro Shinzato, Hideto Yasuda, Takashi Moriya, Masahiro Kashiura, Nobuaki Shime, Keita Morikane.

**Validation:** Yutaro Shinzato, Hideto Yasuda, Masahiro Kashiura.

**Visualization:** Yutaro Shinzato, Hideto Yasuda, Masahiro Kashiura.

**Writing – original draft:** Yutaro Shinzato, Hideto Yasuda, Takashi Moriya, Haruka Taira, Yuki Kishihara, Masahiro Kashiura, Yuki Kotani, Natsuki Kondo, Kosuke Sekine, Nobuaki Shime, Keita Morikane.

**Writing – review & editing:** Yutaro Shinzato, Hideto Yasuda, Takashi Moriya, Haruka Taira, Yuki Kishihara, Masahiro Kashiura, Yuki Kotani, Natsuki Kondo, Kosuke Sekine, Nobuaki Shime, Keita Morikane.

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
