## [Decision Letter · Decision Letter 0]

15 Jul 2024

PONE-D-23-30822Risk factors of phlebitis in patients admitted to the intensive care unit vary according to the duration of catheter dwelling: A post-hoc analysis of the AMOR-VENUS studyPLOS ONE

Dear Dr. Yasuda,

Thank you for submitting your manuscript to PLOS ONE. After careful consideration, we feel that it has merit but does not fully meet PLOS ONE’s publication criteria as it currently stands. Therefore, we invite you to submit a revised version of the manuscript that addresses the points raised during the review process.

We look forward to receiving your revised manuscript.

Kind regards,

Ioannis Savvas, DVM, Ph.D.

Academic Editor

PLOS ONE

4. Please upload a copy of your study protocol that was approved by your ethics committee/IRB as a Supporting Information file. By the study protocol, we mean the complete and detailed plan for the conduct and analysis of the trial approved by the ethics committee/IRB. Please send this in the original language. If this is in a language other than English, please also provide a translation. [https://journals.plos.org/plosone/s/submission-guidelines#loc-guidelines-for-specific-study-types].

Reviewers' comments:

Reviewer's Responses to Questions

**Comments to the Author**

1. Is the manuscript technically sound, and do the data support the conclusions?

Reviewer #1: Partly

Reviewer #2: Yes

Reviewer #3: Yes

2. Has the statistical analysis been performed appropriately and rigorously? 

Reviewer #1: No

Reviewer #2: Yes

Reviewer #3: I Don't Know

3. Have the authors made all data underlying the findings in their manuscript fully available?

Reviewer #1: Yes

Reviewer #2: Yes

Reviewer #3: Yes

4. Is the manuscript presented in an intelligible fashion and written in standard English?

Reviewer #1: Yes

Reviewer #2: Yes

Reviewer #3: Yes

5. Review Comments to the Author

Reviewer #1: Thank you for your informative article.

I have a few questions about the analyses.

line 99: what was the purpose of the analysis of variance and Kruskal-Wallis test?

line 101: what was the purpose of the Fisher exact est of Pearson's chi-square test, as the association of interest is with phlebitis, and this is assessed with Cox regression analyses? Clarify if it is to test for differences according to duration of catheter dwelling group. Should the patient and/or hospital effects be considered in these tests? Should a mixed hierarchical model be better placed to assess these differences?

Is the endpoint really time to phlebitis from insertion of PIVC (rather than phlebitis which is presumably incidence of phlebitis)?

How was patient and ICU accounted for in the Cox regression analyses? Does this need to be a hierarchical model too?

Reviewer #2: Shinzato and colleagues have performed a further post-hoc analysis of this study of phlebitis in 23 ICUs in Japan. Although they report factors that may be associated with phlebitis, there is little justification for the dichotomisation of 24 vs > 72 hours, and I note that many of their findings have been previously reported in their three other papers from this dataset. In particular, the biological plausibility of why certain drugs but not others may be associated with later phlebitis is not well explored, and also the discrepant findings reported here from their previous paper which found a shorter dwell time with phlebitis positive patients vs phlebitis negative patients (Duration of catheter dwell, median (IQR), hour 37.0 (19.2–57.6) with phlebitis vs 44.8 (21.0–81.5) without phlebitis p < 0.01) is also not discussed. How does the confounder of catheter removal in the case of suspected phlebitis (with a commensurate shorter dwell time) affect their findings?

Reviewer #3: The authors conducted a post-hoc study of the AMOR-VENUS study to identify risk factors for PIVC-related phlebitis in intensive care patients, according to catheterization duration. 3348 PIVCs inserted in 1335 patients were included. Several independent variables were identified, depending on the duration of catheter dwell.

Overall, the manuscript is clear and well-written. The present study raises no ethical issues, as the ethical approval of the AMOR-VENUS study included post-hoc analyses.

I am not a statistician. My analysis does not include the statistical methods used by the authors.

Several catheters (2.5 on average) were inserted per patient. How did the authors consider this in their statistical analyses?

Similarly, patients with missing data were excluded from the analysis. How many patients/catheters does this represent?

My main limitation is the implication of the results in clinical practice. The duration of catheterization is rarely known when the catheter is inserted. How the identification of phlebitis risk factors according to catheterization dwell time will change practices deserves to be widely developed in the manuscript.

6. PLOS authors have the option to publish the peer review history of their article (what does this mean? ). If published, this will include your full peer review and any attached files.

**Do you want your identity to be public for this peer review?** For information about this choice, including consent withdrawal, please see our Privacy Policy .

Reviewer #1: No

Reviewer #2: No

Reviewer #3: **Yes: ** Olivier MIMOZ

---

## [Author Response · Author response to Decision Letter 1]

11 Dec 2024

December, 2024

Ioannis Savvas, DVM, Ph.D.

Academic Editor

PLOS ONE

Dear Dr. Savvas,

We are pleased to be given an opportunity to resubmit our manuscript titled "Risk factors of phlebitis in patients admitted to the intensive care unit vary according to the duration of catheter dwelling: A post-hoc analysis of the AMOR-VENUS study" for further consideration in PLOS ONE. The Manuscript ID is: PONE-D-23-30822.

We would like to thank you and the reviewers for the valuable feedback and insightful suggestions, which have helped us improve the quality of our manuscript. I look forward to working with you and the reviewers to move this manuscript closer to publication in the PLOS ONE.

In response to the reviewers' comments, we have made significant revisions, which we believe have strengthened our work. A detailed point-by-point response to each reviewer’s comment is included below. Additionally, the revisions made in the manuscript have been highlighted in the track-changes version.

We hope that the revised manuscript meets your expectations, and we look forward to your editorial decision. Thank you again for considering our work for publication in PLOS ONE.

Sincerely,

Hideto Yasuda

Department of Emergency and Critical Care Medicine,

1-847, Amanuma-cho, Omiya-ku, Saitama City,

Saitama, 330-8503, Japan

Phone: +81-(0)48-647-2111

Fax: +81-(0)48-648-5180

Email: yasudahideto@me.com

Our responses to the Reviewers' comments and suggestions are listed below:

Reviewer #1

1. Comment:

Line 99: What was the purpose of the analysis of variance and Kruskal-Wallis test?

Response:

Thank you for your question. The Kruskal-Wallis test was performed to compare between the three groups in terms of patient background characteristics to assess whether there were significant differences among the groups categorized by catheter insertion time. Our goal was to assess differences in background characteristics that might impact phlebitis risk, rather than to detect pairwise differences between any specific groups.

2. Comment:

Line 101: What was the purpose of the Fisher’s exact test and Pearson’s chi-square test, as the association of interest is with phlebitis, which is assessed with Cox regression analyses? Clarify if it is to test for differences according to duration of catheter dwelling group. Should the patient and/or hospital effects be considered in these tests? Would a mixed hierarchical model be more appropriate for assessing these differences?

Response:

Thank you for pointing this out. The Fisher’s exact test and Pearson’s chi-square test were used for the univariate analysis to detect differences between the groups rather than to explore associations with phlebitis. These tests aimed to provide an initial understanding of background differences among the groups categorized by catheter dwelling duration. Regarding association testing, we used multilevel Cox regression, which incorporated patient and hospital effects as random effects to account for hierarchical variability. Given the recent advances in analysis techniques, it might indeed have been suitable to omit these background tests entirely, as differences in background characteristics have been less frequently tested in recent literature.

3. Comment:

Is the endpoint really time to phlebitis from insertion of PIVC (rather than incidence of phlebitis)?

Response:

Thank you for this query. As you kindly pointed out, the endpoint in our study was "time to phlebitis from insertion of PIVC." Although the exact time of phlebitis onset may not have been precisely recorded, we monitored the catheter insertion site every four hours, allowing us to estimate the time to phlebitis onset with reasonable accuracy.

4. Comment:

How were patient and ICU accounted for in the Cox regression analyses? Should this also be modeled hierarchically?

Response:

Thank you for pointing this out. In our analysis, we used multilevel Cox regression, incorporating both patient and facility (ICU) as random effects to address potential clustering effects. This hierarchical modeling approach allowed us to account for variability in catheter management practices across facilities and among patients who had multiple catheters.

Reviewer #2

1. Comment:

There is little justification for the dichotomization of 24 vs >72 hours.

Response:

Thank you for your comment. We considered categorizing the groups into three distinct categories based on catheter dwelling duration because early infections post-insertion is common in critically ill patients. Although there was no universally accepted cutoff, we selected a 72-hour threshold to achieve a balanced sample size in each group, which could help reduce the risk of overfitting and ensure robust statistical analysis.

2. Comment:

The biological plausibility of why certain drugs but not others may be associated with later phlebitis is not well explored.

Response:

Thank you for your comment. The factors influencing phlebitis onset, particularly medication effects, remain complex and are not fully understood. The existing studies often treat medication as a binary variable (administered or not), which does not capture the timing or dose effects. Ideally, the time-dependent effects of medication on phlebitis onset would require a more sophisticated analysis. In our study, we did not investigate specific timing due to model simplicity; this limitation is noted for future studies.

3. Comment:

the discrepant findings reported here from their previous paper which found a shorter dwell time with phlebitis positive patients vs phlebitis negative patients (Duration of catheter dwell, median (IQR), hour 37.0 (19.2–57.6) with phlebitis vs 44.8 (21.0–81.5) without phlebitis p < 0.01) is also not discussed.

Response:

Thank you for your suggestion. We agree that an explanation was necessary. Accordingly, we have added an explanation to the manuscript (Lines 216-223).

4. Comment:

Discrepant findings regarding dwell time and phlebitis. How does the confounder of catheter removal due to suspected phlebitis (with a corresponding shorter dwell time) affect your findings?

Response:

We appreciate your insightful observation. In our study, early catheter removal due to suspected phlebitis might have led to a shorter dwelling duration, potentially impacting the results. If catheter dwelling duration was reduced due to phlebitis occurrence, this might have influenced the observed relationship between phlebitis incidence and dwelling time. Cases of extended dwelling durations could have shown different phlebitis outcomes. This limitation has been noted in our manuscript, as we recognize the need for further research to clarify this aspect. We have added this meaning to the manuscript (Lines 249-255).

Reviewer #3

1. Comment:

Several catheters (2.5 on average) were inserted per patient. How did the authors consider this in their statistical analyses?

Response:

Thank you for your question. Given the possibility that outcomes might be correlated across multiple catheters in the same patient, we performed a multilevel analysis with both patient and facility factors as random effects. This approach helped us account for the nested structure of our data and reduced the risk of inflated associations due to within-patient correlations.

2. Comment:

Patients with missing data were excluded from the analysis. How many patients/catheters does this represent?

Response:

Thank you for your question. We have added a detailed account of the number of patients and catheters excluded due to missing data. This information is reported in the results section (Line 164-167).

3. Comment:

Implications for clinical practice: The duration of catheterization is rarely known when the catheter is inserted. How will identifying phlebitis risk factors based on dwell time change practices?

Response:

We appreciate your insightful query. Our study identified that risk factors for phlebitis onset varied with catheterization duration. For instance, in cases of high risk for phlebitis within the first 24 hours, scheduled replacement might be considered within that period. Conversely, if no risk factors appeared until after 72 hours, the catheter could potentially be monitored until then, delaying replacement. This approach could help estimate optimal replacement timing, which we discussed in the manuscript (Line 232-242).

We look forward to your feedback and hope that our revised manuscript is now suitable for publication.

---

## [Decision Letter · Decision Letter 1]

3 Jan 2025

PONE-D-23-30822R1Risk factors of phlebitis in patients admitted to the intensive care unit vary according to the duration of catheter dwelling: A post-hoc analysis of the AMOR-VENUS studyPLOS ONE

Dear Dr. Yasuda,

Thank you for submitting your manuscript to PLOS ONE. After careful consideration, we feel that it has merit but does not fully meet PLOS ONE’s publication criteria as it currently stands. Therefore, we invite you to submit a revised version of the manuscript that addresses the points raised during the review process.

We look forward to receiving your revised manuscript.

Kind regards,

Ioannis Savvas, DVM, Ph.D.

Academic Editor

PLOS ONE

Journal Requirements:

Reviewers' comments:

Reviewer's Responses to Questions

**Comments to the Author**

1. If the authors have adequately addressed your comments raised in a previous round of review and you feel that this manuscript is now acceptable for publication, you may indicate that here to bypass the “Comments to the Author” section, enter your conflict of interest statement in the “Confidential to Editor” section, and submit your "Accept" recommendation.

Reviewer #1: (No Response)

Reviewer #3: All comments have been addressed

2. Is the manuscript technically sound, and do the data support the conclusions?

Reviewer #1: Yes

Reviewer #3: Yes

3. Has the statistical analysis been performed appropriately and rigorously? 

Reviewer #1: Yes

Reviewer #3: I Don't Know

4. Have the authors made all data underlying the findings in their manuscript fully available?

Reviewer #1: Yes

Reviewer #3: Yes

5. Is the manuscript presented in an intelligible fashion and written in standard English?

Reviewer #1: Yes

Reviewer #3: Yes

6. Review Comments to the Author

Reviewer #1: There was no text change to address the four points that I made in the comments - thus another reader may experience the same confusion as I. Please add some important words to the manuscript to clarify these points (Reviewer 1)

Reviewer #3: I thank the authors for having taken all my comments into account. The manuscript is now suitable for publication.

7. PLOS authors have the option to publish the peer review history of their article (what does this mean? ). If published, this will include your full peer review and any attached files.

**Do you want your identity to be public for this peer review?** For information about this choice, including consent withdrawal, please see our Privacy Policy .

Reviewer #1: No

Reviewer #3: **Yes: ** Olivier MIMOZ

---

## [Author Response · Author response to Decision Letter 2]

15 Feb 2025

February, 2025

Ioannis Savvas, DVM, Ph.D.

Academic Editor

PLOS ONE

Dear Dr. Savvas,

We are pleased to be given an opportunity to resubmit our manuscript titled "Risk factors of phlebitis in patients admitted to the intensive care unit vary according to the duration of catheter dwelling: A post-hoc analysis of the AMOR-VENUS study" for further consideration in PLOS ONE. The Manuscript ID is: PONE-D-23-30822.

We would like to thank you and the reviewers for the valuable feedback and insightful suggestions, which have helped us improve the quality of our manuscript. I look forward to working with you and the reviewers to move this manuscript closer to publication in the PLOS ONE.

In response to the reviewers' comments, we have made substantial and detailed revisions to comprehensively respond to all comments. Specifically, we have:

• Elaborated on our statistical methodology, explicitly explaining the rationale for each test used.

• Refined the definition of the study endpoint, ensuring clarity regarding how "time to phlebitis" was determined.

• Justified the use of hierarchical modeling, with a clear explanation about why this approach was necessary.

• Expanded the discussion about clinical implications, outlining how our findings could help establish catheter management strategies in intensive care settings.

We sincerely appreciate the reviewers' valuable insights, which have helped us significantly improve our manuscript. For clarity, all revisions have been highlighted in the track-changes version of the manuscript.

We hope that the revised manuscript meets your expectations; we look forward to your editorial decision. Thank you once again for considering our work for publication in PLOS ONE.

Sincerely,

Hideto Yasuda

Department of Emergency and Critical Care Medicine,

1-847, Amanuma-cho, Omiya-ku, Saitama City,

Saitama, 330-8503, Japan

Phone: +81-(0)48-647-2111

Fax: +81-(0)48-648-5180

Email: yasudahideto@me.com

Our responses to the Reviewers' comments and suggestions are listed below:

Reviewer #1

1. Comment:

There was no text change to address the four points that I made in the comments - thus another reader may experience the same confusion as I. Please add some important words to the manuscript to clarify these points.

Response: We thank you for your valuable and insightful feedback. We sincerely apologize for any lack of clarity in our previous revision and have now made the necessary modifications to explicitly respond to all four comments. The following revisions have been made to ensure the manuscript is clearer and more comprehensive.

1. Line 99: What was the purpose of the analysis of variance and Kruskal-Wallis test?

Response:

Thank you for your question. The analyses of variance (ANOVA) and Kruskal-Wallis test were used to compare background characteristics among the three catheter insertion time groups to determine whether significant differences existed. This was necessary for confirming that the groups were comparable before performing further statistical analyses. We have clarified this in the Methods section (Lines 100-102).

2.Line 101: What was the purpose of the Fisher’s exact test and Pearson’s chi-square test, as the association of interest is with phlebitis, which is assessed with Cox regression analyses? Clarify if it is to test for differences according to duration of catheter dwelling group. Should the patient and/or hospital effects be considered in these tests? Would a mixed hierarchical model be more appropriate for assessing these differences?

Response:

Thank you for pointing this out. In the revised manuscript, we have explicitly stated that Fisher’s Exact Test and Pearson’s Chi-Square Test were used solely for preliminary analyzing background characteristics and were not intended to infer causal relationships with phlebitis risk.

Furthermore, we have emphasized that hierarchical modeling (multilevel Cox regression) was applied in the main analysis to appropriately account for both patient- and ICU-level effects. This would ensure that confounding due to clustering at the institutional and patient levels was minimized (Lines 104-108).

3.Is the endpoint really time to phlebitis from insertion of PIVC (rather than incidence of phlebitis)?

Response:

Thank you for your insightful query. We have now explicitly clarified in the Methods section that the study endpoint was "time to phlebitis from insertion of PIVC," as determined based on regular 4-hour monitoring.

This approach ensures that the earliest possible detection of phlebitis onset is made while maintaining consistency in timepoint evaluation across patients (Lines 113-115).

4.How were patient and ICU accounted for in the Cox regression analyses? Should this also be modeled hierarchically?

Response:

Thank you for pointing this out. We used multilevel Cox regression, incorporating both patient and facility (ICU) as random effects to address potential clustering effects. In the revised manuscript, we have now explicitly justified our choice of this approach, ensuring clarity on why hierarchical modeling was necessary (Lines 106-108).

We sincerely appreciate the time and effort invested by the reviewers and editors for evaluating our manuscript. We believe that these revisions have significantly strengthened our study and ensured its clarity, rigor, and clinical relevance.

We look forward to your feedback and hope that our revised manuscript now fully meets the criteria for publication in PLOS ONE.

---

## [Editor Report · Decision Letter 2]

21 Feb 2025

Risk factors of phlebitis in patients admitted to the intensive care unit vary according to the duration of catheter dwelling: A post-hoc analysis of the AMOR-VENUS study

PONE-D-23-30822R2

Dear Dr. Yasuda,

We’re pleased to inform you that your manuscript has been judged scientifically suitable for publication and will be formally accepted for publication once it meets all outstanding technical requirements.

Kind regards,

Ioannis Savvas, DVM, Ph.D.

Academic Editor

PLOS ONE

---

## [Editor Report · Acceptance letter]

PONE-D-23-30822R2

PLOS ONE

Dear Dr. Yasuda,

I'm pleased to inform you that your manuscript has been deemed suitable for publication in PLOS ONE. Congratulations! Your manuscript is now being handed over to our production team.

Kind regards,

on behalf of

Prof. Ioannis Savvas

Academic Editor

PLOS ONE